Characterization of the first vaginal Lactobacillus crispatus genomes isolated in Brazil

http://orcid.org/0000-0002-8557-3865 Oliveira de Almeida Marcelle 1
Carvalho Rodrigo 1
http://orcid.org/0000-0002-1067-1882 Figueira Aburjaile Flavia 1
http://orcid.org/0000-0002-6823-5995 Malcher Miranda Fabio 1
Canário Cerqueira Janaína 1
http://orcid.org/0000-0002-7635-9656 Brenig Bertram 2
http://orcid.org/0000-0003-3880-5886 Ghosh Preetam 3
http://orcid.org/0000-0002-8032-1474 Ramos Rommel 4
Kato Rodrigo Bentes 5
http://orcid.org/0000-0001-7299-3724 de Castro Soares Siomar 6
http://orcid.org/0000-0002-4082-1132 Silva Artur 4
http://orcid.org/0000-0002-4775-2280 Azevedo Vasco 1 vascoariston@gmail.com
Canário Viana Marcus Vinicius 1
1 Department of Genetics, Ecology, and Evolution, Federal University of Minas Gerais , Belo Horizonte, Minas Gerais , Brazil
2 Institute of Veterinary Medicine, University of Göttingen , Göttingen , Germany
3 Department of Computer Science, Virginia Commonwealth University , Richmond, VA , USA
4 Department of Genetics, Federal University of Pará , Belém, Pará , Brazil
5 Post-graduation Program in Bioinformatics, Federal University of Minas Gerais , Belo Horizonte, Minas Gerais , Brazil
6 Department of Immunology, Microbiology, and Parasitology, Federal University of Triângulo Mineiro , Uberaba, Minas Gerais , Brazil
Mora-Montes Hector
Electronic publication date: 2021 Mar 10
Publication date: 2021
Volume: 9
Electronic Location ID: e11079
Received 2020 Dec 9; Accepted 2021 Feb 17
Copyright: © 2021 Oliveira de Almeida et al.
Copyright year: 2021
Copyright holder: Oliveira de Almeida et al.
License: This is an open access article distributed under the terms of the Creative Commons Attribution License, which permits unrestricted use, distribution, reproduction and adaptation in any medium and for any purpose provided that it is properly attributed. For attribution, the original author(s), title, publication source (PeerJ) and either DOI or URL of the article must be cited.
License URL: https://creativecommons.org/licenses/by/4.0/

Keywords: Lactobacillus, Genomics

Funding: Fundação Coordenação de Aperfeiçoamento de Pessoal de Nível Superior (CAPES) Conselho Nacional de Desenvolvimento Científico e Tecnológico (CNPq) Federal University of Minas Gerais (UFMG) Federal University of Pará (UFPA) This work was supported by Fundação Coordenação de Aperfeiçoamento de Pessoal de Nível Superior (CAPES), Conselho Nacional de Desenvolvimento Científico e Tecnológico (CNPq), Federal University of Minas Gerais (UFMG) and Federal University of Pará (UFPA). The funders had no role in study design, data collection and analysis, decision to publish, or preparation of the manuscript.

==============================
Background

Lactobacillus crispatus is the dominant species in the vaginal microbiota associated with health and considered a homeostasis biomarker. Interestingly, some strains are even used as probiotics. However, the genetic mechanisms of L. crispatus involved in the control of the vaginal microbiome and protection against bacterial vaginosis (BV) are not entirely known. To further investigate these mechanisms, we sequenced and characterized the first four L. crispatus genomes from vaginal samples from Brazilian women and used genome-wide association study (GWAS) and comparative analyses to identify genetic mechanisms involved in healthy or BV conditions and selective pressures acting in the vaginal microbiome.

Methods

The four genomes were sequenced, assembled using ten different strategies and automatically annotated. The functional characterization was performed by bioinformatics tools comparing with known probiotic strains. Moreover, it was selected one representative strain (L. crispatus CRI4) for in vitro detection of phages by electron microscopy. Evolutionary analysis, including phylogeny, GWAS and positive selection were performed using 46 public genomes strains representing health and BV conditions.

Results

Genes involved in probiotic effects such as lactic acid production, hydrogen peroxide, bacteriocins, and adhesin were identified. Three hemolysins and putrescine production were predicted, although these features are also present in other probiotic strains. The four genomes presented no plasmids, but 14 known families insertion sequences and several prophages were detected. However, none of the mobile genetic elements contained antimicrobial resistance genes. The genomes harbor a CRISPR-Cas subtype II-A system that is probably inactivated due to fragmentation of the genes csn2 and cas9. No genomic feature was associated with a health condition, perhaps due to its multifactorial characteristic. Five genes were identified as under positive selection, but the selective pressure remains to be discovered. In conclusion, the Brazilian strains investigated in this study present potential protective properties, although in vitro and in vivo studies are required to confirm their efficacy and safety to be considered for human use.

Introduction

The vaginal microbiota of reproductive-age women is classified into at least five types called community state types (CST). Four of them are dominated by Lactobacillus crispatus (CST-I), L. gasseri (CST-II), L. iners (CST-III), or L. jensenii (CST-V). The CST-IV is characterized as having a significantly lower number of lactobacilli and an increased number and diversity of strict and facultative anaerobes (Ravel et al., 2011). The lactobacilli use the glycogen supplied by the host as a carbon source and create a protective environment against infections or colonization by pathogens and non-indigenous microbes by the production of L- and/or D-lactic acid, bacteriocins, hydrogen peroxide, competition for tissue adhesion, enhancement of the protective mucus layer integrity and modulation of the innate immune system response (Ravel et al., 2011; Smith & Ravel, 2017).

CST-IV and III increase the risk of bacterial vaginosis (BV) due to L. iners being less effective in controlling the vaginal microbiota. BV is a condition characterized by a microbiota similar to CST-IV, vaginal pH > 4.5 and production of amino acid compounds, sometimes associated with clinical symptoms including discharge, fishy odor, and presence of clue cells. These conditions damage the host defenses and favor the development of opportunistic microorganisms that behave like pathogens (Smith & Ravel, 2017; Barrientos-Durán et al., 2020). BV can be asymptomatic or associated with gynecological and obstetric complications, besides increasing the risk of sexually transmitted infections (Barrientos-Durán et al., 2020).

CST-I, dominated by L. crispatus, is most associated with vaginal health. This species is considered a biomarker of a healthy microbiota, and some strains are used as a probiotic to treat BV. Its effect as a probiotic is not entirely clear, but it is believed to involve competitive exclusion strategies (Almeida et al., 2019). It was shown to outcompete Gardinerella vaginallis for tissue adhesion in vitro (Ojala et al., 2014) and, as other lactobacilli, inhibit the growth of pathogens in vivo by the production of lactic acid, but not hydrogen peroxide (Tachedjian, O’Hanlon & Ravel, 2018). The decrease in L. crispatus is associated with BV, but the causes are not well understood. The differences in persistence and protection among strains is probably influenced by genetic differences that, if described, could be applied in the screening of more efficient probiotic strains (Almeida et al., 2019).

Comparative genomic analyses have been performed to identify genomic features of L. crispatus associated with female urogenital tract using samples from North America, Europe and Asia (Ojala et al., 2014; Abdelmaksoud et al., 2016; France, Mendes-Soares & Forney, 2016; Van der Veer et al., 2019; Pan, Hidalgo-Cantabrana & Barrangou, 2020; Petit & Read, 2020; Zhang et al., 2020). Analyses that searched for genomic features associated with lactobacilli-dominated (healthy) or BV microbiota were performed using samples isolated from the USA (Abdelmaksoud et al., 2016) and Netherlands (Van der Veer et al., 2019). Those studies point out mechanisms related to persistence during BV, such as phase variation, rather than protection against this infirmity (Van der Veer et al., 2019). In Latin America, although L. crispatus has been previously studied in Brazil as vaginal isolates (Branco et al., 2010) or as part of the vaginal microbiota (Martinez et al., 2008; Marconi et al., 2020), no complete genome has been sequenced, analyzed, and deposited in GenBank.

In this study, we characterize the first L. crispatus genomes isolated in Brazil from healthy vaginal microbiomes and used genome-wide association study (GWAS) and positive selection analyses to identify genetic mechanisms involved healthy or BV conditions and selective pressures acting in the vaginal microbiome.

Materials and Methods

Genome sequencing, assembly, and annotation

In a previous study, vaginal fluid samples were collected from individuals diagnosed as healthy or with BV, with the approval of the ethics committee in research (COEP) of the Federal University of Minas Gerais (protocols ETIC 062/03 and 212/07) (Branco et al., 2010). L. crispatus strains were identified by cellular morphology (Gram-positive bacilli or coccobacilli), biochemistry test (catalase-negative) and 16S-23S rRNA restriction profiling (Branco et al., 2010). The genomes of strains CRI4, CRI8, CRI10 and CRI17 isolated from four healthy patients were sequenced using HiSeq 2500 (Illumina, San Diego, CA, USA) with paired-end libraries of 2 × 150 bp. The sequencing reads quality was examined using FastQC v0.11.8 (Andrews, 2015). The sequencing reads were mapped to the genomes of 46 L. crispatus vaginal isolates to filter out contaminants (Table 1) using bowtie v2 (Langmead & Salzberg, 2012), and the mapped reads were extracted using Samtools v1.7-2 (Li et al., 2009).

Table 1 List of Lactobacillus crispatus genomes.

Strain	Condition	Microbiome	Metadata	Country	GenBank	Reference	
RL02	BV	DVM	BV-positive	Netherlands	NKLR01	Dols et al. (2016), Van der Veer et al. (2019)	
RL07	BV	DVM	BV-positive	Netherlands	NKLN01	Dols et al. (2016), Van der Veer et al. (2019)	
RL13	BV	DVM	BV-positive	Netherlands	NKLI01	Dols et al. (2016), Van der Veer et al. (2019)	
RL14	BV	DVM	BV-positive	Netherlands	NKLH01	Dols et al. (2016), Van der Veer et al. (2019)	
RL15	BV	DVM	BV-positive	Netherlands	NKLG01	Dols et al. (2016), Van der Veer et al. (2019)	
RL17	BV	DVM	BV-positive	Netherlands	NKLE01	Dols et al. (2016), Van der Veer et al. (2019)	
RL19	BV	DVM	BV-positive	Netherlands	NKLD01	Dols et al. (2016), Van der Veer et al. (2019)	
RL20	BV	DVM	BV-positive	Netherlands	NKLC01	Dols et al. (2016), Van der Veer et al. (2019)	
RL21	BV	DVM	BV-positive	Netherlands	NKLB01	Dols et al. (2016), Van der Veer et al. (2019)	
RL23	BV	DVM	BV-positive	Netherlands	NKLA01	Dols et al. (2016), Van der Veer et al. (2019)	
RL24	BV	DVM	BV-positive	Netherlands	NKKZ01	Dols et al. (2016), Van der Veer et al. (2019)	
RL25	BV	DVM	BV-positive	Netherlands	NKKY01	Dols et al. (2016), Van der Veer et al. (2019)	
RL28	BV	DVM	BV-positive	Netherlands	NKKV01	Dols et al. (2016), Van der Veer et al. (2019)	
RL30	BV	DVM	BV-positive	Netherlands	NKKT01	Dols et al. (2016), Van der Veer et al. (2019)	
RL31	BV	DVM	BV-positive	Netherlands	NKKS01	Dols et al. (2016), Van der Veer et al. (2019)	
RL33	BV	DVM	BV-positive	Netherlands	NKKQ01	Dols et al. (2016), Van der Veer et al. (2019)	
VMC1	BV	DVM	History of BV. <50% lactobacilli and >50% of BV-associated taxa	USA	LJCZ01	Abdelmaksoud et al. (2016)	
VMC2	BV	DVM	History of BV. <50% lactobacilli and >50% of BV-associated taxa	USA	LJDA01	Abdelmaksoud et al. (2016)	
VMC3	BV	DVM	History of BV. <50% lactobacilli and >50% of BV-associated taxa	USA	LJGP01	Abdelmaksoud et al. (2016)	
VMC4	BV	DVM	History of BV. ~86% L. crispatus and ~12% BV-associated taxa	USA	LJGQ01	Abdelmaksoud et al. (2016)	
CRI4	Healthy	–	Healthy	Brazil	JABERN01	This study	
CRI8	Healthy	–	Healthy	Brazil	JABERO01	This study	
CRI10	Healthy	–	Healthy	Brazil	JABERP01	This study	
CRI17	Healthy	–	Healthy	Brazil	JABERQ01	This study	
2029	Healthy	–	Healthy, probiotic strain	Russia	AVFH2	Abramov et al. (2014)	
125-2-CHN	Healthy	–	Healthy	China	ACPV01	Ojala et al. (2014), www.beiresources.org	
AB70	Healthy	–	Healthy	South Korea	CP026503, CP026504	Chang et al. (2019)	
CIP 104459	Healthy	–	Healthy	France	VOMA01	Clabaut et al. (2020)	
CTV-05	Healthy	–	Healthy, probiotic strain	–	ADML01	Hemmerling et al. (2010), Ojala et al. (2014)	
JV-V01	Healthy	–	Normal human vaginal flora	–	ACKR01	Witkin et al. (2013), Ojala et al. (2014), www.beiresources.org	
MV-1A-US	Healthy	–	Healthy	USA	ACOG02	Witkin et al. (2013), Ojala et al. (2014), www.beiresources.org	
MV-3A-US	Healthy	–	Healthy	USA	ACQC01	Witkin et al. (2013), Ojala et al. (2014), www.beiresources.org	
RL03	Healthy	LVM	BV-negative	Netherlands	NKLQ01	Dols et al. (2016), Van der Veer et al. (2019)	
RL05	Healthy	LVM	BV-negative	Netherlands	NKLP01	Dols et al. (2016), Van der Veer et al. (2019)	
RL06	Healthy	LVM	BV-negative	Netherlands	NKLO01	Dols et al. (2016), Van der Veer et al. (2019)	
RL08	Healthy	LVM	BV-negative	Netherlands	NKLM01	Dols et al. (2016), Van der Veer et al. (2019)	
RL09	Healthy	LVM	BV-negative	Netherlands	NKLL01	Dols et al. (2016), Van der Veer et al. (2019)	
RL10	Healthy	LVM	BV-negative	Netherlands	NKLK01	Dols et al. (2016), Van der Veer et al. (2019)	
RL11	Healthy	LVM	BV-negative	Netherlands	NKLJ01	Dols et al. (2016), Van der Veer et al. (2019)	
RL16	Healthy	LVM	BV-negative	Netherlands	NKLF01	Dols et al. (2016), Van der Veer et al. (2019)	
RL26	Healthy	LVM	BV-negative	Netherlands	NKKX01	Dols et al. (2016), Van der Veer et al. (2019)	
RL27	Healthy	LVM	BV-negative	Netherlands	NKKW01	Dols et al. (2016), Van der Veer et al. (2019)	
RL29	Healthy	LVM	BV-negative	Netherlands	NKKU01	Dols et al. (2016), Van der Veer et al. (2019)	
RL32	Healthy	LVM	BV-negative	Netherlands	NKKR01	Dols et al. (2016), Van der Veer et al. (2019)	
SJ-3C-US	Healthy	–	Healthy	Iran	ADDT01	Eslami et al. (2016)	
V4	Healthy	–	Healthy	France	SRLG01	Clabaut et al. (2019)	
VMC5	Healthy	LVM	No history of BV. >90 % L. crispatus and <10 % BV-associated bacterial taxa	USA	LJOK01	Abdelmaksoud et al. (2016)	
VMC6	Healthy	LVM	No history of BV. Dominated by L. crispatus and L. jensenii	USA	LJOL01	Abdelmaksoud et al. (2016)	
VMC7	Healthy	LVM	No history of BV. >90 % L. crispatus and <10 % BV-associated bacterial taxa	USA	LJOM01	Abdelmaksoud et al. (2016)	
VMC8	Healthy	LVM	No history of BV. >90 % L. crispatus and <10 % BV-associated bacterial taxa	USA	LJON01	Abdelmaksoud et al. (2016)	
Note:

LVM, Lactobacilli-dominated vaginal microbiota; DVM, Dysbiotic vaginal microbiota.

Ten different assemblies were generated for each genome using SPAdes v3.14.0 (Bankevich et al., 2012) (assembly 1), Unicycler v0.4.5 (Wick et al., 2017) with SPAdes v3.14.0 (2), ABySS v2.0 (Simpson et al., 2009) (3), MaSuRCA v3.4.0 (Zimin et al., 2013) (4) and an in house pipeline (https://github.com/engbiopct/assembly-hiseq) that generates six other assemblies (5–10). In the in-house pipeline, six strategies combine different software. The best k-mer values were identified using KmerStream v1.1 (Melsted & Halldórsson, 2014). Adapters were removed using AdapterRemoval v2.3.1 (Schubert, Lindgreen & Orlando, 2016). The genome assemblers were Edena v3.131028 (Hernandez et al., 2008) and SPAdes v3.13.0 (Bankevich et al., 2012). The six assembly strategies were: Edena (5), KmerStream and SPAdes (6), KmerStream and SPAdes, using Edena assembly as trusted contigs (7), AdapterRemoval, KmerStream and SPAdes (8), reads processed by AdapterRemoval and the raw reads as input, KmerStream, and SPAdes, using Edena assembly as trusted contigs (9), and AdapterRemoval, KmerStream, and SPAdes, using Edena assembly as trusted contigs (10).

The best assembly for each genome was determined using QUAST v5.0.2 (Gurevich et al., 2013). Then, the genome’s paired-read sequencing data was used for scaffolding with SSPACE v3.0 (Boetzer et al., 2011) and contig extension and gap filling with GapFiller v1.1.1 (Boetzer & Pirovano, 2012). Finally, more gaps were closed using contigs from the other nine assemblies and the chromosome of L. crispatus strain AB70 (CP026503.1) (Chang et al., 2019) as a reference, using GFinisher (Guizelini et al., 2016). The presence of plasmids was investigated using PlasmidFinder 2.1 (Carattoli et al., 2014). The four genomes were identified from the species L. crispatus using the Type (Strain) Genome Server (Meier-Kolthoff & Göker, 2019). The assemblies completeness was evaluated using BUSCO v4.0.6 (Seppey, Manni & Zdobnov, 2019), based on the presence of 402 single-copy orthologous genes shared within Lactobacillales. The genomes were annotated using Prokka v1.11 (Seemann, 2014). The GenBank accession numbers of the genomes from strains CRI4, CRI8, CRI10 and CRI17, isolated from healthy patients, are JABERN01, JABERO01, JABERP01 and JABERQ01, respectively (Table 2). Type Strain Genome Server (Meier-Kolthoff & Göker, 2019) was used to confirm the taxonomic classification of the 50 samples as L. crispatus strains.

Table 2 Statistics of genome sequencing, assembly, and annotation of the Lactobacillus crispatus strains CRI4, CRI8, CRI10, and CRI17.

	CRI4	CRI8	CRI10	CRI17	
SRA accession	SRR13201099	SRR13201098	SRR13201097	SRR13201096	
Replicon accession	JABERN01	JABERO01	JABERP01	JABERQ01	
Completeness (%)	99.5	99.2	99.0	99.0	
Size (bp)	2,376,268	2,330,310	2,418,420	2,384,332	
Contig	100	65	65	79	
N50 (bp)	44.691	69.313	75.243	58.032	
L50	19	10	11	14	
CDS	2.438	2.329	2.478	2.393	
Plasmid	None	None	None	None	

Criteria of public genomes selection

We included in the analysis genomes available in public databases using the following criteria: (i) vaginal isolate, (ii) the metadata explicitly informs the health condition of the individual, and/or the microbiome classification. A total of 46 samples were classified as belonging to “healthy” (26) or “BV” (20) condition groups and were used in previous studies (Hemmerling et al., 2010; Ojala et al., 2014; Abramov et al., 2014; Abdelmaksoud et al., 2016; Eslami et al., 2016; Dols et al., 2016; Chang et al., 2019; Clabaut et al., 2019, 2020; Van der Veer et al., 2019). Table 1 shows the genomes list, including the microbiome classification, when available, and the terms in metadata used to classify the sample.

Probiotic features

Bacteriocins and linear azol(in)e-containing peptides (LAPs) were predicted using BAGEL4 (Van Heel et al., 2018). The enzymes involved in the production of L- and D-lactate and hydrogen peroxide were identified using KEGG Mapper/BlastKOALA, under the pyruvate metabolism pathway. Adhesins were predicted using eggNOG-mapper v2 (Huerta-Cepas et al., 2017). Protein IDs were identified using BLASTp (Camacho et al., 2009) using the GenBank non-redundant (nr) database, selecting hits with 100% identity and coverage. Pathways involved in the biosynthesis of antimicrobial drugs with clinical importance (Chokesajjawatee et al., 2020) were predicted using KEGG Mapper.

Safety assessment of the strains for probiotic applications

Detection of plasmids, insertion sequences, prophages, and CRISPR-Cas elements was performed using PlasmidFinder (Carattoli et al., 2014), ISEScan (Xie & Tang, 2017), PHASTER (Arndt et al., 2016), and CRISPRCasFinder (https://crisprcas.i2bc.paris-saclay.fr/CrisprCasFinder/Index), respectively. The domains of the identified Cas proteins were predicted using InterProScan (Jones et al., 2014) and NCBI’s Conserved Domain Database (Lu et al., 2020). Multiple alignments of sequences of interest were performed using the Clustal Omega web service (Larkin et al., 2007). Local alignment of sequences of interest across genomes was performed using BLASTn (Camacho et al., 2009) implemented in PATRIC (Davis et al., 2020).

Virulence factor genes were detected using the databases VFDB (Liu et al., 2019) and Ecoli_VF (https://github.com/phac-nml/ecoli_vf), while antimicrobial resistance genes were detected using the databases ARG-ANNOT (Gupta et al., 2014), CARD (Alcock et al., 2020), MEGARes (Doster et al., 2019), NCBI AMRFinderPlus (Feldgarden et al., 2019) and ResFinder (Bortolaia et al., 2020). The screening using these databases was performed using ABRicate (https://github.com/tseemann/abricate) with default parameters.

The presence of the toxins hemolysins, enzymes involved in the synthesis of biogenic amines and other undesirable genes, as listed by Chokesajjawatee et al. (2020), were manually screened using KEGG Mapper (Kanehisa & Sato, 2020) implemented in BlastKOALA v2.2 (Kanehisa, Sato & Morishima, 2016).

In vitro detection of phages

One representative strain (L. crispatus CRI4) was subjected to the induction of lysogenic bacteriophages according to previous studies (Kiliç et al., 1996; Raya & H’bert, 2009). A total of 200 µL of the culture was initially inoculated in 5 mL of Man–Rogosa–Sharpe medium. The optical density (OD) was measured until reaching DO600 = 0.2. Then, 0.4 µg/mL of Mitomycin C (Sigma, St. Louis, MO, USA) was added in the culture and incubated at 37 °C overnight. Afterward, the supernatant was collected and filtered in sterile 0.22 µm membrane. A 10 µL sample of the filtered lysate was applied to a 200 mesh grid at the UFMG electron microscopy center (CM-UFMG) and visualization was performed in Tecnai G2-12 Transmission Electron Microscope, SpiritBiotwin FEI, 120 kV.

Comparative analyses using public genomes

We performed comparative analysis including the for strains from Brazil and 46 public genomes (Table 1) to reconstruct their phylogeny and to identify adaptations that could be related to the colonization of the vaginal niche by testing for association between gene presence/absence and features of interest, and detection of positive selection in protein-coding genes.

For phylogenomic analysis, L. helveticus DSM20075 genome (GenBank accession ACLM01) was used as an outgroup, the conserved genes across all 51 genomes were estimated by Roary v3.6.0, and their nucleotide sequences were aligned using MAFFT (Katoh et al., 2005) implemented in Roary. The alignment was used as input for IQ-Tree v1.6.12 (Nguyen et al., 2015) for phylogenetic inference using the Maximum Likelihood. The confidence values were estimated using 1,000 rounds of bootstrapping. The tree was edited using iTOL (Letunic & Bork, 2019).

We tested the associations of gene presence/absence with a health condition and geographical locations suggested by the phylogenetic tree. A GWAS based on gene presence/absence was performed using Scoary v1.6.16 (Brynildsrud et al., 2016). Scoary estimates association by pairwise comparisons on a phylogeny (Maddison, 2000) to correct population structure and permutation. The input was a gene presence-absence matrix from the 50 L. crispatus genomes estimated using Roary and a phylogenetic tree generated by IQ-Tree v1.6.12, utilizing the core gene alignment calculated by Roary, and a matrix containing the presence-absence of the features across the samples.

A genome-scale positive selection analysis was performed using POTION v1.1.2 (Hongo et al., 2015). To generate the input, FastOrtho (https://github.com/PATRIC3/FastOrtho) was used to identify ortholog groups across the 50 L. crispatus genomes. The file containing the orthologous group’s information and multifasta files containing nucleotide sequences of protein-coding genes were used as input for POTION v1.1.2. The genome-scale positive selection analysis used site tests with the models M1 and M2, and M7 and M8 (Yang & Nielsen, 2002). The POTION configuration file is available as Data S1. The function of the identified proteins was annotated using eggNOG-mapper (Huerta-Cepas et al., 2017), the subcellular localization using SufG+ v1.2.1 (Barinov et al., 2009), and the GenBank protein ID using BLASTp (Agarwala et al., 2016).

Results

Genome sequencing and taxonomy

The four sequenced genomes were assembled as drafts, and no plasmid was found. Table 2 shows the statistics of genome assembly and annotation. The genomes completeness ranged from 99% to 99.5%, while the reference strain AB70, available as a complete genome sequenced using PacBio RS II platform (Chang et al., 2019), had its completeness estimated as 99%. The four sequenced genomes and the 46 public genomes were classified as L. crispatus by TYGS, with dDDH > 70% and G+C content divergence of less than 1% to the strain JCM 1185T (Data S2).

Probiotic features

Features associated with probiotic effects in the four strains from Brazil are shown in Table 3. We identified genes involved in the biosynthesis of D-lactate (1), L-lactate (3), hydrogen peroxide (1), bacteriocins (9), LAPs (1) and adhesins (10, five classes) across the four genomes. No pathway for biosynthesis of antimicrobial drugs of clinical importance was found.

Table 3 Genomic features related to probiotic effects in Lactobacillus crispatus strains CRI4, CRI8, CRI10 and CR17.

				Strain	
Feature	Product (Gene)	KEGG ID	Protein ID	CRI4	CRI8	CRI10	CRI17	
Lactic acid synthesis								
	D-lactate			1	1	1	1	
	D-lactate dehydrogenase [EC:1.1.1.28] (ldhA)	K03778	WP_005720611	1	1	1	1	
	L-lactate			3	3	3	3	
	L-lactate dehydrogenase [EC:1.1.1.27] (ldh)	K00016	WP_005721100	1	1	1	1	
		K00016	WP_005720302	1	WP_170080485	1	WP_005721074	
	L-2-hydroxyisocaproate dehydrogenase (hicDH)	K00016	WP_005727148	WP_005719855	1	1	1	
Bacteriocin								
	Class: 210.2; SakT_alpha			2	2	3	2	
	ggmotif; ComC; Bacteriocin_IIc;	–	WP_005721006	1	1	1	1	
	ComC; L_biotic_typeA; Bacteriocin_IIc; 20.2; bacteriocin_LS2chaina		WP_005721005	1	1	1	1	
	ComC; L_biotic_typeA; Bacteriocin_IIc;	–	WP_005720990	–	–	1	–	
	Class: 70.3; Helveticin-J			1	2	2	2	
	70.3; Helveticin-J	–	WP_005729773	–	1	1	1	
		–	WP_005720754	1	1	1	1	
	Class: 64.3; Enterolysin_A			1	2	2	2	
	64.3; Enterolysin_A	–	WP_005728076	–	1	1	1	
		–	WP_005719715	1	1	1	1	
	Class: 163.2; Penocin_A			–	–	2	1	
	bacteriocinII; Bacteriocin_II; ComC; Bacteriocin_IIc; 163.2; Penocin_A	–	WP_005723822	–	–	1	1	
	Bacteriocin_IIc;	–	WP_005727428	–	–	1	–	
	Class: 6.3; Bacteriocin_helveticin_J			1	1	1	1	
	6.3; Bacteriocin_helveticin_J	–	WP_005728268	WP_005718134	1	1	1	
	Class: LAPs			-	-	1	-	
	Putative nitr oreductase MJ1384	–	WP_005721909	–	–	1	–	
Hydrogen peroxide synthesis								
	Pyruvate oxidase [EC:1.2.3.3] (poxL)	K00158	WP_005723618	1	1	1	1	
Adhesin								
	Putative adhesin		WP_005728236	1	1	1	1	
			WP_005729490	–	1	1	1	
Antimicrobial production	–	–	–	–	–	–	–	

Safety assessment of the strains for probiotic applications

Features associated with safety in the four strains from Brazil are shown in Table 4. About mobile elements, no plasmid was predicted, as previously stated. A total of 131 to 184 IS from 14 known families were identified in the four genomes, and one new family was predicted across CRI8 (2 copies), CRI10 (2) and CRI17 (1) (Data S3). The new IS has a size of 2,088 bp and harbors two genes coding a site-specific integrase-resolvase (IS607-like family, GenBank protein ID WP_005728427.1) and a transposase (IS605 family, AZR15009.1) (Data S4). The multiple alignments of the five copies show the IS is fragmented in the contig 44 of strain CRI10, and the difference among the complete sequences is a SNP (T to G) in position 1,350 (Data S5). A BLASTn against the four strains from Brazil and other 123 public L crispatus genomes database identified hits with ≥99% identity and ≥98% coverage with 42 genomes from strains isolated from the female human urogenital tract, 39 of them public genomes (Data S3).

Table 4 Genomic features related to safety in Lactobacillus crispatus strains CRI4, CRI8, CRI10, and CR17.

				Strain	
Feature	Product (Gene)	KEGG ID	BLASTp hit	CRI4	CRI8	CRI10	CRI17	
Mobile elements								
Plasmid				–	–	–	–	
Insertion sequences				131	171	178	184	
Prophages				2 questionable, 4 incomplete	5 incomplete	1 questionable, 5 incomplete	1 questionable, 4 incomplete	
CRISPR-Cas system				CAS-TypeIIA	CAS-TypeIIA	CAS-TypeIIA, CAS-TypeIIC	CAS-TypeIIA	
Bacterial toxins								
	Hemolysin A (tlyA)	K06442	WP_005723149	–	1	1	1	
	Putative hemolysin (tlyC)	K03699	WP_005727867	1	1	1	1	
	Hemolysin-III related (hlyIII)	K11068	WP_005720215	1	1	1	1	
Bile salt deconjugation								
	Choloylglycine hydrolase/bile salt hydrolase (cbh)	K01442	WP_005718943	1	1	1	1	
Biogenic amine formation								
	Ornithine decarboxylase (odcI)	K01581	WP_005727730	1	1	1	1	
Antimicrobial resistance	–	–	–	–	–	–	–	

Five or six prophages were predicted across the four genomes, classified as questionable (score 70–90) or incomplete (score < 70), with most of them close to contig ends. In strains CRI8 and CRI17, some of the predicted prophages harbored the new IS (Fig. 1; Data S6). We identified CRISPR-Cas systems subtype II-A in the four strains and subtype II-C only in strain CRI10 (Table 4; Data S7). A closer examination of the subtype II-A system reveals that all four genomes contain CRISPR loci and Cas proteins-encoding genes cns2, cas2, cas1, and cas9. However, a transposase is inserted in csn2 in each strain, resulting in two gene fragments. The gene cas9 is also fragmented in all strains, with a transposase inserted between fragments in strains CRI4 and CRI8. CRI17 has an extra copy of cas9. The subtype II-C system is represented in CRI10 by a single cas9 gene (Fig. 2). All the cas9 CDSs lack one or more domains.

Figure 1 Insertion sequence from IS607-like family located in two prophages (regions) in Lactobacillus crispatus strain CRI8.

(A) Region 2 and (B) Region 3. The insertion sequences from the IS607-like family are two subsequent transposases, located in region 2 and at the end of region 3. Att, Attachment Site; Coa, Coat protein (purple); Hyp, Hypothetical protein (green); Int, Integrase (blue); Pla, Plate protein (orange); PLP, Phage-like Protein (cyan); Tra, Transposase (olive).

Figure 2 CRISPR-Cas systems subtype II-A predicted in four Lactobacillus crispatus strains from Brazil.

(A) Strain CRI4, (B) CRI8, (C) CRI10 and (D) CR17. The genes csn2 and cas9 are fragmented in the four genomes. Colors of the protein coding sequences green—csn2, cas2 and cas1; red—cas9 fragments; pink—transposase.

No virulence or antimicrobial resistance gene was predicted using ABRicate. With the search for virulence and undesirable genes using KEGG Mapper/BlastKOALA, we identified two hemolysins (tlyC and hlyIII), one bile salt hydrolase (cbh), and one enzyme involved in the biosynthesis of putrescine (Ornithine decarboxylase, odcI). Manual screening of Prokka annotation showed a third hemolysin, Hemolysin A. The search for toxin biosynthesis identified an “Ornithine decarboxylase” (odcI) (WP_005727730.1) involved in the synthesis of the biogenic amine putrescine, in all for strains.

In vitro detection of phages

Several rod-shaped particles of varying length and thickness were observed by electron microscopy, some of them similar to Myoviridae bacteriophages. This pattern was repeated throughout the slide (Fig. 3).

Figure 3 Transmission electron microscopy of the strain CRI4 filtrate.

(A) Presence of stick-shaped structures. (B) Structure similar to a capsid with contractile tail (Myoviridae).

Comparative analyses using public genomes

The strains did not cluster according to a health condition. However, strains from Brazil were the closest to strains from the Netherlands, even in different clusters (Fig. 4). The GWAS based on gene presence/absence implemented in Scoary did not identify an association with a health condition (p > 0.05). Due to the clustering of strains from Brazil and the Netherlands, we tested the association with these two geographical locations and the result was also negative. A total of 8 protein-coding genes were identified as under positive selection (q < 0.05). After manual curation for false positives caused by alignment artifacts, five genes were obtained: a surface exposed “S-layer protein precursor,” three phage related proteins, and a hypothetical protein (Table 5; Data S8).

Table 5 Genes under positive selection in Lactobacillus crispatus genomes from healthy and bacterial vaginosis samples.

Product (Gene)	COG	Location	Sequences/genomes	PS sites	Positions	Protein ID	
Integrase core domain protein	L	CYTOPLASMIC	53/50	1	160	EKB63650	
S-layer protein precursor	S	SECRETED	42/50	1	130	EKB61518	
Hypothetical protein	–	CYTOPLASMIC	15/50	11	31, 39, 49, 48, 52, 58, 94, 95, 103, 107, 126	WP_126708926	
ORF6N domain protein	K	CYTOPLASMIC	10/50	2	240, 256	WP_133463822	
Tyrosine recombinase XerS (xerS)	L	CYTOPLASMIC	4/50	1	231	WP_133475995	
Note:

PS, positively selected sites.

Figure 4 Phylogenomic tree of Lactobacillus crispatus strains from vaginal isolates.

The tree was built using the nucleotide sequences of 118 core genes predicted by Roary and aligned by MAFFT, 1,000 rounds of bootstrapping, and Maximum Likelihood phylogenetic inference implemented in IQ-TREE. Filled squares—bacterial vaginosis samples, Empty squares—samples from healthy individuals.

Discussion

In the first four L. crispatus sequenced genomes from Brazil, we identified genes involved in the protective properties of vaginal lactobacilli (Smith & Ravel, 2017) such as the production of lactic acid, bacteriocins, hydrogen peroxide and adhesion (Table 3).

For safety reasons, a probiotic strain should not present features associated with virulence and antimicrobial resistance, as well as mobile elements that could transfer those features to other microorganisms from the host microbiome (Pariza et al., 2015; Chokesajjawatee et al., 2020). No antimicrobial resistance gene was found, and only three hemolysins were predicted as virulence factors, which have also been previously identified in other Lactobacillus probiotic strains (Zafar & Saier, 2020) (Table 4). The hemolysin-III is widespread across Lactobacillus species, including strains considered as safe and commercially available (Chokesajjawatee et al., 2020). This result implies these genes should not be a safety concern. However, the hemolytic activity of those proteins from Lactobacillus must be verified in further studies. We predicted the synthesis of the toxic biogenic amine putrescine by the ornithine decarboxylase pathway in all four strains (Table 4). This is one of the pathways in which the decarboxylation of amino acids and organic acids generates a proton motive force that can regulate intracellular pH (acidic stress response) and generate ATP (Romano et al., 2014; Del Rio et al., 2018). The production of biogenic amines is an important safety issue when screening probiotic strains as they may cause intoxication when consumed in high concentrations (García-Villar, Hernández-Cassou & Saurina, 2009; Linares et al., 2011).

Concerning mobile elements, as expected for probiotic strains, no plasmids were found. Despite of numerous ISs from 14 known families (Table 4; Data S3) and one from a new family-specific of vaginal isolates (Datas S3, S4, and S5), none of them were associated with antibiotic resistance genes.

Four to six incomplete prophages across the genomes were revealed (Table 4; Data S6). Some incomplete prophages could be the result of an assembly artifact, as the genomes were assembled as drafts and most prophages were located close to a contig end. The possible role of the newly described IS, and the associated prophages in the genome evolution (Siguier, Gourbeyre & Chandler, 2014) or adaptation to the urogenital niche should be investigated. Moreover, the presence of complete phages was confirmed by electron microscopy in CRI4 lysate after Mitomycin C induction (Fig. 3). Therefore, our results suggest the possibility of lysogeny of this strain.

The presence of a functional CRISPR-Cas system could prevent the infection by bacteriophages (Crawley et al., 2018) that could influence the vaginal microbiota (Macklaim et al., 2013). We detected the subtype II-A in the four genomes and II-C in CRI10 (Table 4; Data S7). However, all csn2 and cas9 genes across the genomes are fragmented (Fig. 2), lacking one or more domains. Also, the subtype II-C in CRI10 is probably a prediction artifact, as a single subtype with two sets of cas genes can be detected as multiple subtypes (Crawley et al., 2018) and only the subtypes I-B, I-E, and II-A were reported for this species (Petit & Read, 2020). The result suggests that the CRISPR-Cas systems are not functional due to gene fragmentation which could compromise the prevention of phage infections.

In the phylogenomic tree, health-related condition does not show clustering (Fig. 4), a result also found in other studies (Abdelmaksoud et al., 2016; Pan, Hidalgo-Cantabrana & Barrangou, 2020). However, Brazil-Netherlands clusters were formed. The GWAS performed using Scoary did not find gene presence/absence associated with the health condition or the Brazil-Netherlands clusters. This result is different from a previous study, which has shown an association of transposases and a glycosyltransferase being more abundant in BV strains (Van der Veer et al., 2019). The differences obtained between association studies could be due to the multifactorial characteristic of BV (Barrientos-Durán et al., 2020; Marconi et al., 2020).

The positive selection analysis identified five genes (Table 5; Data S8). The S-layer precursor has the surface layer A protein (SLAP) domain described in slpB from L. acidophilus ATCC 4356 (Boot, Kolen & Pouwels, 1995). S-layer proteins form symmetric, porous, lattice-like layers that cover the cell surface with poorly known functions that can involve mediation of bacterial adherence to host cells, extracellular matrix proteins, or protective or enzymatic functions (Hynönen & Palva, 2013). Surface exposed proteins are located in the interface with the environment and can be under positive selection due to interaction with several factors such as antimicrobial compounds, viruses, hosts, and other bacteria (Petersen et al., 2007). The three phage related proteins have domains for DNA biding, integration, and recombination. The selective pressures acting in those proteins have yet to be identified.

Conclusions

The first L. crispatus genomes from vaginal isolates from Brazil presented several genes associated with probiotic characteristics. Although mobile genetic elements were detected, they do not present antimicrobial resistance genes that could be transmitted to other bacteria. For safety issues, the functionality of the hemolysin related genes must be further experimentally confirmed. No genomic feature was associated with healthy and BV conditions, and the positive selection was predicted in an S-layer protein and phage related genes but have yet to be investigated.

Supplemental Information

Supplemental Information 1 Configuration file for POTION pipeline.

Click here for additional data file.

Supplemental Information 2 Taxonomic classification of 50 Lactobacillus crispatus genomes.

Click here for additional data file.

Supplemental Information 3 Insertion sequences in the Lactobacillus crispatus genomes isolated in Brazil.

Click here for additional data file.

Supplemental Information 4 New insertion sequence from IS607-like family predicted in Lactobacillus crispatus CRI8, CRI10, and CR17.

Shows the complete sequence in strain CRI8.

Click here for additional data file.

Supplemental Information 5 Multiple alignment of the new insertion sequence in the Lactobacillus crispatus genomes isolated in Brazil.

Click here for additional data file.

Supplemental Information 6 Prophages predicted in the Lactobacillus crispatus genomes isolated in Brazil.

Click here for additional data file.

Supplemental Information 7 CRISPR-Cas system elements predicted in the Lactobacillus crispatus genomes isolated in Brazil.

Click here for additional data file.

Supplemental Information 8 Genome-scale positive selection analysis of Lactobacillus crispatus genomes using site models.

Click here for additional data file.

Additional Information and Declarations

Competing Interests

Author Contributions

DNA Deposition

Data Availability

Rommel Ramos and Vasco Azevedo are Academic Editors for PeerJ.

Marcelle Oliveira de Almeida conceived and designed the experiments, performed the experiments, analyzed the data, prepared figures and/or tables, authored or reviewed drafts of the paper, and approved the final draft.

Rodrigo Carvalho conceived and designed the experiments, performed the experiments, analyzed the data, prepared figures and/or tables, authored or reviewed drafts of the paper, and approved the final draft.

Flavia Figueira Aburjaile conceived and designed the experiments, performed the experiments, analyzed the data, authored or reviewed drafts of the paper, and approved the final draft.

Fabio Malcher Miranda performed the experiments, authored or reviewed drafts of the paper, and approved the final draft.

Janaína Canário Cerqueira performed the experiments, analyzed the data, prepared figures and/or tables, authored or reviewed drafts of the paper, and approved the final draft.

Bertram Brenig performed the experiments, authored or reviewed drafts of the paper, genome sequencing, and approved the final draft.

Preetam Ghosh analyzed the data, authored or reviewed drafts of the paper, and approved the final draft.

Rommel Ramos conceived and designed the experiments, performed the experiments, authored or reviewed drafts of the paper, and approved the final draft.

Rodrigo Bentes Kato performed the experiments, analyzed the data, authored or reviewed drafts of the paper, and approved the final draft.

Siomar de Castro Soares conceived and designed the experiments, authored or reviewed drafts of the paper, and approved the final draft.

Artur Silva conceived and designed the experiments, authored or reviewed drafts of the paper, and approved the final draft.

Vasco Azevedo conceived and designed the experiments, authored or reviewed drafts of the paper, and approved the final draft.

Marcus Vinicius Canário Viana conceived and designed the experiments, performed the experiments, analyzed the data, prepared figures and/or tables, authored or reviewed drafts of the paper, and approved the final draft.

The following information was supplied regarding the deposition of DNA sequences:

The genome sequences described here are available at GenBank: JABERN01, JABERO01, JABERP01 and JABERQ01.

The following information was supplied regarding data availability:

The genome sequencing reads are available at NCBI SRA: SRR13201099, SRR13201098, SRR13201097 and SRR13201096.

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
