# Peer review of "Characterization of the first vaginal Lactobacillus crispatus genomes isolated in Brazil"

_PeerJ, doi:10.7717/peerj.11079_

## Round 0.1 · original submission · Minor Revisions

Two reviewers have assessed your manuscript and provided positive comments about your work. Please address all of them in a revised version of your work.

·

Basic reporting

The work presented by Marcelle Oliveira de Almeida and colleagues reports the sequencing of four strains previously isolated from individuals diagnosed as healthy or with bacterial vaginosis. Overall, the strains show genomic features that suggest can be safe for further use as a complementary therapy for bacterial vaginosis. Genomic analysis indicates that no antibiotic resistance genes are present in any of the four strains analyzed. Strains harbor Cas enzymes but they may also be inactive. Prophages are also present and may be active. Metabolic coding genes analysis shows that these strains may render protective benefits.
Overall, the manuscript is well written, methods and experimental design are sound, and the conclusions can be drawn from the data obtained.

Experimental design

This reviewer finds the experimental design sound and well-conducted. My only request regarding this is that authors add the accession numbers of the datasets deposited in GenBank and please add the statement the source of each strain as “from healthy patient” or “BV patient”.

Validity of the findings

In the general comments section, the authors will find the concerns rose by this reviewer and some suggestions that I think need to be addressed. The findings reported in this manuscript are relevant, although the prophage issue for this reviewer is important to complement this manuscript. Some analysis seems poorly discussed and that is highlighted in the General comments section. Overall, this reviewer thinks that the manuscript is of worth and can be published, but some modifications are needed to strengthen the manuscript.

Additional comments

Major comments:
In the methods section please add a statement of the GenBank data accession number, although given in Table 1, this is a must statement either at the end of the manuscript as data availability or in the methods section.
This reviewer finds that the phylogenomic tree is not fully discussed. I agree that the clustering between healthy and BV samples is not present but is interesting that the sequenced strains in this study belong to a cluster mostly represented by samples from the Netherlands. Does this have any meaning? Is there any relevant feature found in those strains? Also, most of the samples from the Netherlands are from BV samples, if the sequences from these samples are separately analyzed with the Brazil samples, any interesting feature surfaces? I suggest adding a line in the discussion section regarding this feature.
Line 308-310, to further support the safety of these strains, I encourage authors to provide at least as a supplementary figure the results for lytic phages present in CR14 and CRI17 strains, perhaps a plaque assay. This reviewer understands that authors may be preparing a report of these findings, but due to the statements in this manuscript regarding the safety of these strains, I encourage authors to provide the information requested, please read lines 332-333 in this regard.
Although interesting, the search for CRISPR loci and Cas coding genes seems out of place from the scope of the paper, I suggest authors revise the discussion (lines 311-322) to explain why this is important to BV and microbiome dynamics. Also, the authors emphasize that the CRISPR-Cas systems are not functional in these strains, I encourage authors to provide a line why this is relevant in their study and provide a line regarding the reading frame of these genes that fully explain that may be inactive.
Finally, regarding the probiotic effects of these strains, is there synteny between the lactic acid synthesis enzymes? Genomic rearrangements are found between strains?
Minot comments:
Please use in the abstract bacterial vaginosis instead of BV.
Line 69, please correct to “healthy microbiota”
Line 83 please correct to “BV microbiota were performed using samples isolated from the USA…”
Line 99 please change to: “with the approval of the ethics…”
In the methods section, please deposit or provide a legend to the in-house pipeline generated so readers can refer to the supplementary information provided and perhaps deposit the scripts used in a GNU or similar agreements.
Line 274: please correct to “health-related condition does not show clustering (Fig 1) as…”
Line 286, please correct to “a safety concern”
Line 302, please change to “carry a second gene, a …”
In line 345 please add a reference for the risk factors associated with vaginal homeostasis and dysbiosis.
Figure 1 legend, please correct to Filled squares, bacterial vaginosis samples, Empty squares, samples from healthy individuals.
In Table 4 the Prophages identified, please correct to questionable in full.

Reviewer 2 ·

Basic reporting

The text is mostly clear and unambiguous. I have noted several small grammatical errors that are completely understandable as the authors are not native English speakers. The issues listed below just illustrate these errors (I've only listed those in the Discussion, there are others throughout that can be easily fixed by a native speaker).
Several minor grammatical errors are present in the manuscript:
Line 295: Fourteen known IS families and one new family (were) was predicted
Line 301: IS (ISs?) from families…. carry (a) second gene…
Line 312: ‘Type II and II-A are the most common type, and subtype in Lactobacillus, respectively, and…’ Is it rather ‘most common type and subtype in Lactobacillus, respectively…?
Line 331: ‘suggest that they attend the safety requirements (replace attend with ‘bear’ or something similar)?
Line 333: lysogeny (not lisogeny)
Line 361: change ‘has’ to ‘have’
Article structure is fine. I found figure 2 to be not so important and could be deleted or removed to supplementary.

Experimental design

The work is within the scope of the journal. It is not completely clear why an additional four genomes of L. crispatus are necessary, but the authors emphasize the need for coverage in South America. Hence, they publish these four sequences. The data is good, methods are well described.

Validity of the findings

Overall, this paper meets these criteria. I found the GWAS analysis the least useful and felt it could be deleted,

Additional comments

Suggest a proof by a native English speaker to correct minor grammatical errors. Overall, there is no concern about the data. The assemblies are good, if not completely closed. Some of the gaps could be associated with the broken genes identified and that could be examined more carefully. Clearly, the results are not 'earth-shattering', but should be relevant for people in the field. In addition to the previous comments, there are several minor issues:

1. Line 239: is it a mistake that subtype II-C Crispr system is only in CRI10? It seems to be present in CRI17 in Table 4?

2. Fig. 3. A legend with the colors identified would be better than the description (green, pink, red?).

3. Positive selection: one of the 8 protein genes was identified as secreted (was this identified with appropriate targeting sequences)? How was ‘cytoplasmic’ determined (lack of targeting sequence?).

4. Line 306-307: ‘ four to six incomplete prophages …. the genomes were not closed’. Were the phage genomes split by gaps in the assemblies?

I might suggest abbreviating the Discussion as it is rather extensive, and in particular, the GWAS section (see above comment).

---

## Round 0.2 · accepted · Accept

The authors have addressed all the points raised by the reviewers.